# Why do self-referent cues facilitate mathematical word problem-solving? Insights from eye tracking

Joshua J. March[1]*, Janet F. McLean[2], Josephine Ross[3], Jan R. Kuipers[4], Sheila J. Cunningham[2]

1 Department of Psychological Sciences & Health, University of Strathclyde, Glasgow, United Kingdom, 2 Division of Psychology & Forensic Sciences, Abertay University, Dundee, United Kingdom, 3 Psychology, School of Humanities, Social Sciences & Law, University of Dundee, Dundee, United Kingdom, 4 Psychology, Faculty of Natural Sciences, University of Stirling, Stirling, United Kingdom

* joshua.march@strath.ac.uk

## Abstract

Associating information with the self enhances processing of that information, with simple text cues like self-referent pronouns (i.e., 'You') increasing response speed and accuracy in processing tasks. Research suggests this can be applied in educational contexts, such as children's mathematical word problem-solving. Whilst children show faster and more accurate word problem-solving when self-pronouns are included in the text, the mechanisms underlying these effects are unclear. The current study extends previous research by using eye-tracking to monitor 9- to 11-year-old children's processing during mathematical word problem-solving. Children were faster to solve subtraction problems that contained a self-referential pronoun, but this was not the case for addition problems. Eye tracking data revealed that faster processing time was driven by reduced fixation length on referent information in the self-pronoun problems across problem types: children spent less time looking at self-pronouns than terms referring to another person (e.g., character names). This suggests that self-pronouns may facilitate problem-solving by supporting active storage of items bound to self in working memory, reducing the need for revisitation during mathematical word-problem solving. This is likely to be particularly beneficial for more cognitively challenging problems, providing an explanation for patterns of self-reference effects reported previously.

## 1. Introduction

Both children and adults show cognitive processing biases for information relating to the self, which tends to be preferentially attended to and remembered [1–3]. Self-referent stimuli such as one's own name [4,5] and face [6,7] can evoke these biases, but they are also elicited by items associated with the self temporarily [8–11],

**Data availability statement:** The current study was pre-registered on the AsPredicted online repository in May 2022 (anonymous version: https://aspredicted.org/C7B_NC8). Files of all stimuli, analysis scripts and data can be found on AsPredicted's associated open data repository ResearchBox (https://researchbox.org/2612).

**Funding:** This work was funded by research grant ES/T000465/1 from the Economic and Social Research Council (ESRC). The funders had no role in study design, data collection and analysis, decision to publish, or preparation of the manuscript.

**Competing interests:** The authors have declared that no competing interests exist.

supporting the robustness of the effects. In addition, self-processing biases can occur in the absence of explicit association training; simply including self-relevant pronouns like 'I' or 'You' in task materials can be enough to produce enhanced performance [12,13].

An important application of self-processing biases may be in educational settings, where self-cues could be incorporated to facilitate learners' task performance. For example, presenting written text in the first- or second-person perspective could represent a low-cost, easily implemented method of supporting processing. This reasoning was tested by D'Ailly and colleagues [12,14] who investigated the effects of including 'You' in numerical word problems. D'Ailly et al. [12] tested 7- to 11-year-old children on problems involving addition and subtraction operations (e.g., "*John has 5 marbles. Peter has 2 marbles more than John. How many marbles does Peter have?*"). When one of the character names was replaced with the word 'You', participants responded more quickly and accurately to the problem. Cunningham et al. [15] recently replicated this effect using addition and subtraction problems that varied by wording difficulty. They found that the inclusion of a self-pronoun reduced children's mathematical word problem-solving time and increased their accuracy, with accuracy effects driven primarily by self-advantages in the more difficult conditions (i.e., subtraction problems with 'inconsistent' wording, where the operation needed was not immediately obvious from the problem wording). This finding suggests that self-pronoun inclusion may be particularly effective in classroom tasks with a high working memory load.

The effects of self-pronouns on mathematical word-problem solving are argued to be underpinned by self-processing biases in attention and working memory [15]. Word problem solving has a high working memory load because details such as numbers, characters, and operations must be kept active and manipulated during problem resolution [16]. This load puts storage and processing demands on the attention system, which may be alleviated by self-cues. Self-cues are stored more reliably than other stimuli even in short-term working memory, perhaps due to their ability to bind item representations together, increasing overall storage capacity [17–19]. Self-cues are identified very early in visual attention and capture attention automatically (for a review, see Humphrey and Sui [1]), reducing attention allocation demands and thus potentially freeing resources for storage and processing. Long-term self-associated cues may also elicit facilitatory mechanisms related to the self-concept, which is easily accessible (in a near constant state of activation) and requires little effort to store or retrieve in working memory relative to other constructs [17,20]. By increasing the capacity of working memory to keep problem elements active, self-cues may reduce executive demands and increase the fluency of problem solving.

This mechanism may explain the tendency for self-cues problems to be solved more quickly [12,14,15]. Easier binding of problem elements in the creation of the situational model [21,22] may facilitate storage of the problem information in working memory for self-referent trials, reducing the need to repeatedly revisit numbers and characters while performing the operation (cf. [23]). For example, in numerical word problems, self-related information (e.g., 'you have two apples') may be more

effectively bound together as a single unit than other-related information (e.g., Johnny takes one apple), making this component of the situational model easier to process. This is consistent with eye tracking findings with young adults [24] and 10- to 13-year-old children [25] which suggest that successful mathematical word problem-solving relies on constructing a shared mental representation uniting text comprehension and numerical calculation. Alternatively, when word problems include a self-pronoun, children may spend less time processing the whole problem due to the appearance of the self in the problem naturally increasing their attentional focus or engagement with the question set. In this case, although there may be no difference in the relative visual attention afforded to the content of the self and other-referent information within the question, numerical word problem-solving as a whole may be faster. To detangle these explanations for decreased response time to self-referent questions, the current study will use eye-tracking to examine the time children spend reading different elements of numerical word problems that either include or omit self-pronouns.

Eye-tracking technology has been used widely in mathematical processing research (for review see [26,27]), with the duration of gaze fixation on a particular location generally interpreted as a measure of cognitive processing of the stimulus presented at that location [28,29]. Eye-tracking research has been applied in mathematical word-problem solving in particular, with patterns of fixation duration revealing numerical processing patterns such as complexity effects [30–32], the order of operand processing [33], and spatial orienting to number lines [28]. The pattern of reading and re-reading specific elements of a problem can also be revealing; Moeller et al. [34] showed that when calculations involved carrying digits in double-digit addition, total fixation time (attributed to re-visitations) was longer on the 10-digit, to which the carry-on calculation was to be applied. Similarly, Hegarty et al. [23] found that when wording was more linguistically complex, participants showed more repeated word fixations, suggesting forward-and-backwards scanning of the information. For example, 'inconsistent' word problems include relational terms that are incompatible with the required mathematical operation (e.g., addition problems containing the term "less"). Thus, eye-tracking technology can be used to infer the elements of a word problem that require particular attention, not just at a visual but at an executive level.

To apply eye-tracking methodology to the study of self-cue response facilitation in word-problem solving, we adapted Cunningham et al.'s [15] tasks, examining how the presence of a self-cue (i.e., "You") affected 9- to 11-year-old children's eye fixation on different elements of numerical word problems. As we were primarily interested in the fluidity of processing and problem solving, our numerical problems were designed to be relatively easy to solve accurately, with response time and eye tracking serving as key dependent variables. To examine how attention to specific elements of the problem might vary across self-referent and non self-referent problems, we compared fixation duration on specific words (e.g., referent terms, nouns, operands and operation terms) across referents and operation (addition versus subtraction). Specific directional hypotheses regarding eye-fixation times were not preregistered due to the exploratory nature of the study. However, based on previous research, children may be expected to be faster when responding to problems including self-referential pronouns (i.e., "You") than problems that refer to another person (e.g., "Eve"), particularly under more challenging processing conditions, such as subtraction problems [15]. This might be associated with differences in visual attention, such that children may have shorter total fixation on the self-referent pronouns relative to the other-referent pronouns within the maths problem, facilitating response time. Alternatively, children may have shorter response times for self-referent problems due to more global facilitated processing (the self draws attention and increases engagement in classroom tasks), showing no difference in the amount of visual attention given to self and other referent pronouns within the question.

## 2. Method

### 2.1 Transparency and openness

The current study was pre-registered on the AsPredicted online repository in May 2022 (anonymous version: https://aspredicted.org/C7B_NC8). The sample size, data exclusions, manipulation, and measures are all reported below. Files of all stimuli, analysis scripts and data can be found on AsPredicted's associated open data repository ResearchBox

(https://researchbox.org/2612). The data were analysed in R [35] and R Studio version 4.1.2. [36] with the tidyverse [37], tidyr [38], dplyr [39], readl and readr [40], psych [41], labelled [42], moments [43], rstatix [44], lme4 [45], broom [46], broom.mixed [47], data.table [48], lubridate [49], slider [50], emmeans [51], and Hmisc [52] packages.

## 2.2 Participants

We recruited children in P5, P6, and P7 classes from four urban primary schools across Scotland between 1st May and 30th September 2022. Written consent was obtained from local council authorities when requested, and permission was obtained from headteachers to distribute consent forms electronically to parents/caregivers. Only children whose care-givers provided written electronic consent via Qualtrics were invited to take part in the study. Express verbal consent was required for every participant. The study was approved by Abertay University's Department of Sociological and Psychological Sciences ethics committee (project approval number EMS5713).

The initial sample included 57 participants but one participant's data was not recorded due to experimenter error, leaving a sample of 56 participants (31 female, $M_{Age}$ = 125.6 months, $SD_{Age}$ = 8.71 months, ranging from 108 to 147 months). As detailed in our pre-registration, previous studies looking at self-referencing in maths problems suggest that the effect of Referent is associated with a partial $\eta^2$ effect size between.13 and.16. Taking the more conservative estimate (.13), a power analysis in MorePower 6.0 [53] for a 2 * 2 repeated-measures design with one 2-level effect of interest (i.e., Referent) and $\alpha$ = .05 suggests a sample size between 56 and 90 participants (for when power is.8 or.95). We are therefore at the lower acceptable limit for our sample size.

## 2.3 Materials and apparatus

Participants completed 32 maths questions, 16 addition and 16 subtraction, further divided into eight questions with a Self Referent (i.e., "You have…") and eight with an Other Referent ("Eve/Zak has…") questions (see the ResearchBox for all experimental materials, https://researchbox.org/2612). The task was administered using the E-Prime 3.0 software in conjunction with the E-Prime Extensions for Tobii [54]. We used a Tobii Pro X3-120 eye tracker [55], which has a sampling rate of 120 Hz (i.e., it records eye position roughly every 8 milliseconds), to record children's eye movements.

We used four maths problem templates, two addition types and two subtraction types (see Table 1), each requiring participants to manipulate two numbers. All word problems were either two or three sentences long. The initial number varied from 3 to 15, the second number varied from 2 to 11, and the answers varied between 3 and 15. Each word problem contained a key noun that was controlled for frequency and familiarity (see below). All word problems with their key nouns, numbers and answers are provided in the online Research Box.

The word problems were counterbalanced for gender, with the Other Referent and the additional names in the problems controlled to avoid matching the participants' stated gender. Thus female participants saw word problems using the Other Referent 'Zak' and that only included male first names. Male participants saw problems using the Other Referent 'Eve' and female first names. 'Zak' and 'Eve' were chosen as they were among the least common 3-letter names out of the 100 most popular names in Scotland from the years 2009–2012 [56], making it likely that participants would have heard the names before, but without running the risk that many participants would be called this. All other names used

**Table 1. Examples of the types of word problems.**

| Operation | Example |
|---|---|
| Addition | You have 5 cups. Gary has 8 cups more than you. How many cups does Gary have? |
| | Zak carried 9 beans to the garden and Troy carried 2. How many beans were carried to the garden? |
| Subtraction | You have 11 bags. Joe has 4 bags less than you. How many bags does Joe have? |
| | Zak had 13 rolls but gave some to Ollie. Zak now has 8 rolls. How many did he give Ollie? |

in the problems were between 3 and 5 letters in length to be similar in length to the Test Referents (i.e., pronoun "you", "Zak/ Eve"), and varied across the problems to avoid familiarity, so only the Test Referents were presented repeatedly. We counterbalanced which word problems were associated with each referent (Self vs Other) between participants. Trial order was randomised in E-Prime.

Because word familiarity and length are important variables affecting fixation duration [57], we controlled for word length and familiarity. We used the ZIPF measure of frequency from the SUBTLEX-UK database [58]. The ZIPF is a standardised metric on a logarithmic scale and is the log10 (word's frequency per billion words) calculated from 201.3 million words from 45099 BBC broadcasts (only using adult TV broadcasts). In the word problems, the key nouns in each sentence ranged from 3.55 to 4.48, with a mean ZIPF of 4.13 (which is between low- and high-frequency). All word problems contained between 17 and 20 words – each word was presented in its own text box in the E-Prime slide. Word length was kept between 1 (for the numbers) and 7 letters, placed in text boxes offset at 6% to 9% screen width.

## 2.4 Procedure

Participants were tested individually by an experimenter in a quiet area of their schools. The experiment was presented on a 31*17.5 cm laptop screen, tilted at 30 degrees and placed at 60 cm from the participant. All testing took place in well-lit rooms but with no direct light sources above or behind the participants. Participants were told that they would be completing some simple maths questions and that their eye movements would be tracked during the activity. Participants were asked to keep still during the task, and not to look away from the screen unless they were taking a break or they had a question. Participants' heads were left unsecured throughout the task. The experimenter performed the 5-point calibration routine provided in the E-Prime Extension for the Tobii package set. After this, instructions appeared on the screen explaining what participants would be asked to do in the maths task, and the experimenter also verbally relayed the task instructions to participants to ensure they understood. Participants then saw a practice question and entered their response before pressing Enter. Participants were instructed that all responses had to be numbers instead of written words and that no words were needed. A final set of instructions then appeared, explaining the task a final time to participants, and the experimenter reminded participants of the rules.

To make sure participants were attending to the screen during the task, at the start of each trial, participants first saw a fixation cross appearing 6 cm from the top of the screen in the centre. Participants had to look continuously at the fixation cross for 1 second before the question appeared in the middle of the screen (see Fig 1). The fixation cross was placed at the top of the screen to avoid overlapping with part of the word problem. After the question appeared, participants read the question and then entered their responses in a 6*2 cm answer box below the questions. Participants typed their responses and pressed Enter to move on to the next trial. No feedback was given to participants between trials. Participants completed 16 trials before being given the option of taking a break for no more than 2 minutes before seeing the remaining 16 questions. Two participants took a break, with the rest completing all 32 questions without a break.

## 2.5 Analysis

The dependent variables for the analyses at trial level were accuracy, response times and word fixation times. *Accuracy* was defined as a binomial variable for each trial, 0 for incorrect and 1 for correct. Correct answers were entered into E-Prime for automatic scoring. *Response time* was defined as the delay in ms between the appearance of the question and a participant's initial key press on the keyboard. Although previous research has focussed on response times for correct trials only [12,15], we were interested in the fluidity of word problem-solving in a naturalistic classroom context, and so included both correctly and incorrectly solved trials. The latter were relatively rare, given that the material was ecologically pitched to the age of the learner. In keeping with our pre-registration, we calculated the natural log for response times (RTs) and used this as the dependent variable in a linear mixed model analysis.

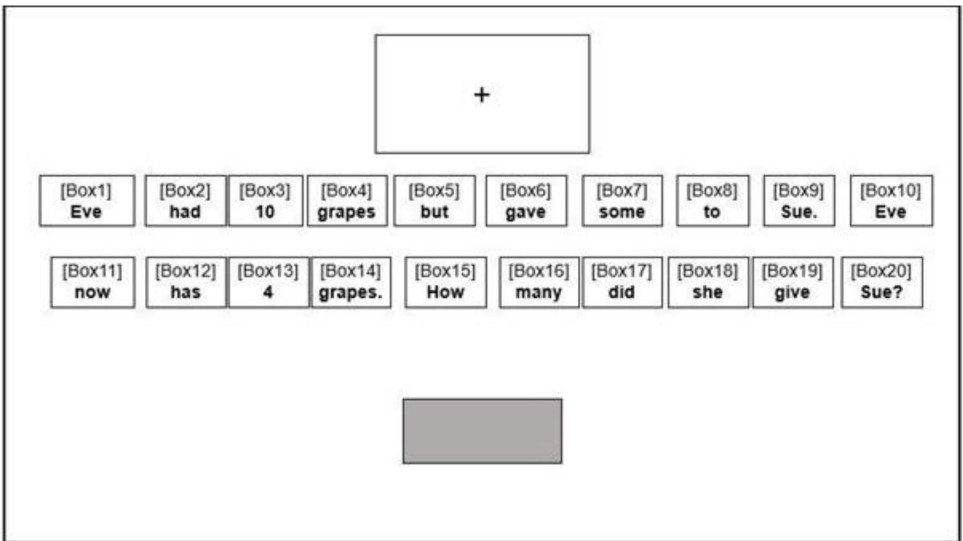

**Fig 1. Visual representation of the experimental set-up in E-Prime.** The fixation cross appeared above the question to avoid overlapping with the word boxes. All word boxes were invisible to the participants. Participants entered their responses into the grey box using the keyboard.

In both the RT and accuracy analyses, we started with the maximal model possible [59], and iteratively removed components, starting with correlations between the random effects, then the by-item slopes for components accounting for the least variance, then the by-subject slopes accounting for the least variance, until the model converged [45]. Treatment contrasts were set for the fixed effects using the bobyqa optimizer. R automatically selects one level as the reference in alphabetical order, so the reference level for Operation Type was Addition, and it was Other for the Referent variable.

The maximal model used for RTs was:

$$\text{lmer(DV} \sim \text{Referent} * \text{Operation} + (1 + \text{Referent} * \text{Operation}|\text{Participant}) + (1 + \text{Referent} * \text{Operation}|\text{Item})$$

The analysis on accuracy used the glmer command due to the categorical nature of the dependent variable.

In our pre-registration, we stated that we would analyse the inverse efficiency score (IES) if the conditions for its use were met. Bruyer and Brysbaert [60]recommend using the IES only when a) the number of errors in the data is lower than 15% and b) there is a high correlation between reaction times and the proportion of errors. While the percentage of incorrect trials in the final sample was only 9.93%, the positive correlation between response time and accuracy was only weak to moderate in strength, $R(51) = .33$, $p = .016$. As such, we forego analysing the IES and focus on analysing the natural response times.

In keeping with our pre-registration, we conducted exploratory analyses to determine how the presence of a self-referential pronoun impacted children's gaze duration on the different word types included in the problems. We grouped all words into six meaningful categories: Test Referent (i.e., self or other), Control Referent (i.e., the other 3rd person), Nouns, Numbers, Key Terms, Non-key Terms. In our pre-registration, we planned to perform analyses on first and average fixation durations. However, our data revealed a substantial amount of unreliable first fixation durations, likely due to the high noise associated with the developmental context, which would overly skew average fixation duration measures. We therefore focussed our analysis on the total fixation time on each word category during a trial, which includes each revisitation. The Tobii Pro X3-120 samples eye gaze information at a rate of 120 Hz, i.e., once every 8 ms. The longer a participant looks at a word, the more hits the eye tracker records for that word. For every trial, we therefore multiplied the

fixation count for each word by 8 to get the total fixation time looking at that word (in ms). The words were each placed in their own region of interest which were separated by approximately 3.1 degrees of visual angle, depending on the viewing distance, while the eye tracker has a resolution of 0.4 degrees.

## 3. Results

### 3.1 Data cleaning

The initial sample included 56 participants. In accordance with our pre-registration, we first removed data for participants with accuracy scores more than 2.5 $SD$s away from the grand mean, resulting in three participants (accuracy scores: 0, 4 and 6 out of 32, respectively) being removed. We then removed any individual trials with response times longer than 2.5 $SD$s above the grand mean ($n = 67$ trials). Finally, we also removed trials for which the total fixation time on the word problem was $< 1000$ ms ($n = 75$ trials). This meant that we removed any trial where participants had looked at the maths problem for less than 1 second (to account for participants skipping trials/inattention). This left 1541 trials for 53 participants (30 female, $M_{Age} = 125.4$ months, $SD_{Age} = 8.58$ months, from 108 to 147 months), which results in slightly less power than the conservative power estimate (for partial $\eta^2 = .13$, $\beta = .8$ if $n = 56$), but sufficient power with the less conservative estimate (for partial $\eta^2 = .16$, $\beta = .8$ if $n = 44$). Of these 1541 trials, 1397 (90.07%) were correct trials.

### 3.2 Behavioural data: Accuracy

Participants scored highly on both types of questions (Addition: $M = 13.6$, $SD = 3.34$; Subtraction: $M = 12.76$, $SD = 4.44$), with only 15 of the 53 participants having accuracy rates below 90% (see Table 2).

The first model that avoided a singular fit contained by-subject and by-item random intercepts, and by-subject slopes for Referent and Operation and by-item slopes for Referent (the fixed effects were all dummy coded, aka treatment coded, with Addition set as the reference category for Operation and Other set as the reference category for Referent):

$$\text{glmer(DV} \sim \text{ Referent} * \text{Operation } + \text{ (1 + Referent } + \text{ Operation|Participant) } + \text{ (1 } + \text{ Referent|Item)}$$

Following Meteyard and Davis [61] we provide a table including the estimates calculated for our fixed effects, their 95% CIs, the $t$-as-$z$ value and associated $p$ value, and the random effects variance (see Table 3). The results showed that none of the effects were significant, all $p$s $> .17$, which is likely due to the near-ceiling performance in these tasks. The ceiling effects were particularly pronounced in Addition questions, with 61.5% of participants having 100% accuracy across referent conditions (mean accuracy 91.9%). In Subtraction questions, a lower but still substantial 44.2% of participants had 100% accuracy (mean accuracy 86.5%). This suggests that while subtraction questions presented more of a challenge to participants, ceiling effects limited the sensitivity of the accuracy measure across operations.

### 3.3 Behavioural data: Response times

Following the outlier trimming discussed above, response times (RTs) ranged between 2.72 seconds and 37.16 seconds, $M = 13.04$ seconds ($SD = 7.14$) (see Table 4).

**Table 2. Average accuracy scores by Operation and Referent (SD in brackets).**

|  | Self Referent | Other Referent |
|---|---|---|
| Addition | 6.89 (1.7) | 6.72 (1.82) |
| Subtraction | 6.48 (2.2) | 6.4 (2.25) |

**Table 3. Estimates of the final linear mixed model for Accuracy. Final _p_ values calculated using the t-as-z method.**

**Fixed effects**

| | Est/Beta | SE | 95% CI | | t | p |
|---|---|---|---|---|---|---|
| **Intercept** | 4.31 | .632 | 3.071–5.55 | | 6.82 | <.001 |
| **Referent** | −.377 | .539 | −1.432 −.679 | | −.699 | .485 |
| **Operation** | −.997 | .73 | −2.427 −.433 | | −1.37 | .172 |
| **Referent * Operation** | .656 | .612 | −.543–1.855 | | 1.07 | .283 |

**Random effects**

| | Variance | | SD | |
|---|---|---|---|---|
| **Participant (intercept) + slopes for Referent and Operation** | .295<br>6.279 | | .54<br>2.506 | |
| **Item (intercept) + slopes for Referent** | .154 | | .393 | |

**Table 4. Average response times in seconds by Operation and Referent (SD in brackets).**

| | Self Referent | Other Referent |
|---|---|---|
| Addition | 12.51 (6.58) | 12.66 (6.83) |
| Subtraction | 13.1 (7.42) | 13.91 (7.63) |

The model that avoided singular fit only contained by-subject and by-item random intercepts.

$$\text{lmer(DV} \sim \text{Referent} * \text{Operation} + \text{(1|Participant)} + \text{(1|Item)}$$

The results show a significant interaction between Referent and Operation (see Table 5).

We followed up the interaction with pairwise comparisons using the emmeans function. The only significant comparison was for the Subtraction questions, showing that children were significantly faster when responding to Subtraction problems, including Self Referents than Other Referents, $t(1458.7) = −3.23$, $p = .007$. Fig 2 illustrates the effect, with data presented in mean response times rather than natural logs for ease of comprehension.

### 3.4 Eye-tracking data: Fixation time per word type

Table 3 displays the total fixation times by each word category (Table 6).

**Table 5. Estimates of the final linear mixed model for Response Times. Final _p_ values calculated using the t-as-z method.**

**Fixed effects**

| | Est/Beta | SE | 95% CI | | t | p |
|---|---|---|---|---|---|---|
| **Intercept** | 9.31 | .068 | 3.07–5.55 | | 137 | |
| **Referent** | .009 | .024 | −1.43 −.679 | | .369 | .712 |
| **Operation** | .031 | .052 | −2.43 −.433 | | .597 | .55 |
| **Referent * Operation** | .069 | .033 | −.542–1.86 | | 2.03 | .042 |

**Random effects**

| | Variance | | SD | |
|---|---|---|---|---|
| **Participant (intercept)** | .173 | | .416 | |
| **Item (intercept)** | .017 | | .131 | |

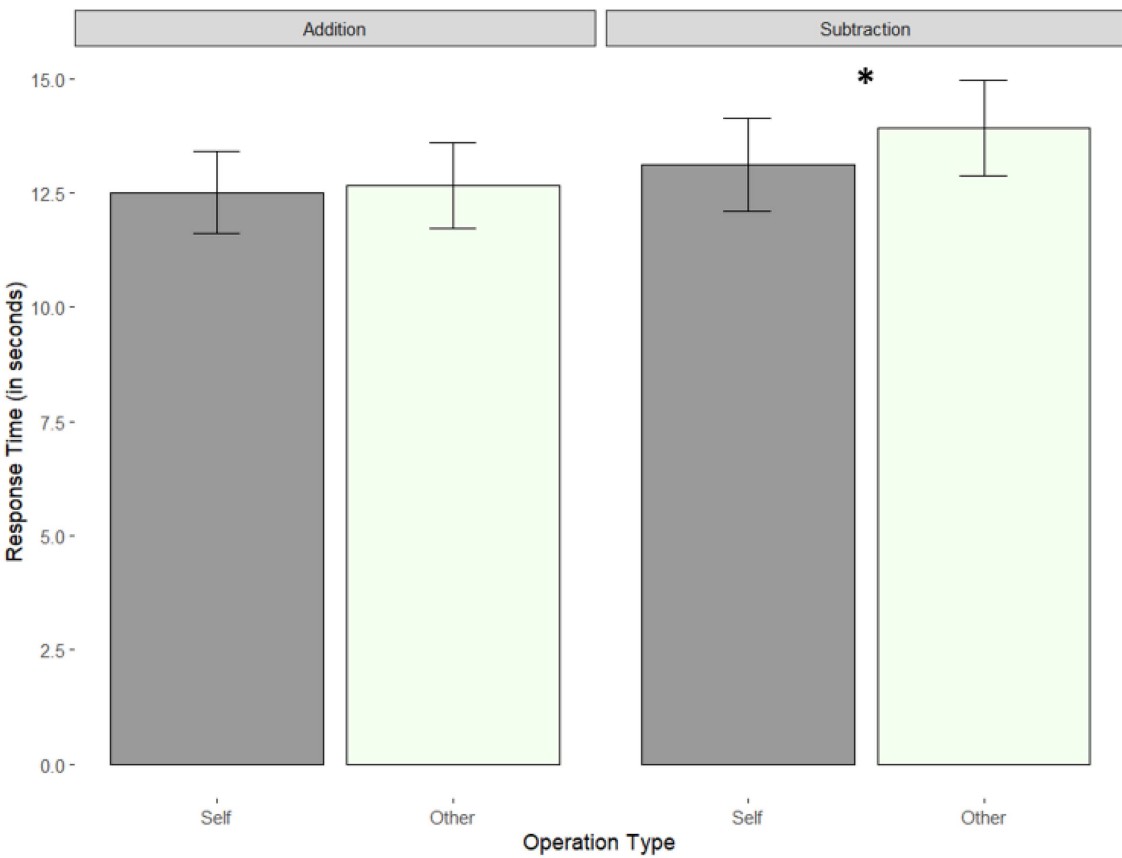

**Fig 2. Response times by Referent and Operation Type (bars indicate SE).**

**Table 6. Total fixation times by word type.**

| Word Type | Example | Total ms fixation time (SE) | Proportional fixation time (%) | Number of observations |
|---|---|---|---|---|
| Target Referent | "You" (self)<br>"Zak/ Eve" (other) | 432 (57) | 5.47 | 1260 |
| Control Referent | "Lucy"<br>"Gary" | 574 (76) | 7.4 | 1445 |
| Noun | "grapes"<br>"ropes" | 1050 (117) | 14.4 | 1521 |
| Number | "13"<br>"8" | 1338 (194) | 17 | 1513 |
| Key Term | "more"<br>"less" | 453 (61) | 6.1 | 654 |
| Non-key term | "has"<br>"some" | 1802 (211) | 23.8 | 1533 |

We conducted linear mixed model analyses to explore the effects of Referent and Operation on the fixation times for each word type, exploring Total fixation time per word type as the DV. Given the exploratory nature of the analysis, we conducted separate analyses for the six word types within each DV. As multiple analyses were being performed, we

applied a Bonferroni correction to our significance cut-off so results for the following tests were considered a result to be significant when $p < .008$ (0.05/ 6).

We again proceeded by attempting the maximal model with the full random effects structure. If that did not converge or if it produced a singular fit, we iteratively removed random effects until the model converged and avoided a singular fit. All resulting models omitted random slopes and only contained by-item and by-participant random intercepts.

This analysis revealed that for Total fixation times, children spent less time looking at the Target Referents in the Addition questions ($M = 381$ ms, $SE = 50$) than in the Subtraction questions ($M = 479$ ms, $SE = 62$), $t = 2.98$, $p = .006$). Regardless of problem type, they also looked at Self-referent terms (i.e., 'You'; $M = 393$ ms, $SE = 51$) for less time than the Other-referent terms (e.g., 'Eve', $M = 470$ ms, $SE = 62$), $t = 4.43$, $p < .001$ (Tables 7 and 8 and Fig 3). These results show that aside from Target Referents, no other word types elicited significantly different fixation times across referent or operation and

**Table 7. Results of linear mixed models for Total Fixation Times by word type. \*\* $p < .005$, \* $p < .05$, uncorrected.**

| Word Category | Fixed Effect | t | p |
|---|---|---|---|
| Target Referent | Referent | 3.25 | .001** |
| | Operation | 2.78 | .006* |
| | Referent * Operation | −0.28 | .854 |
| Control Referent | Referent | −0.47 | .640 |
| | Operation | 1.04 | .300 |
| | Referent * Operation | 1.04 | .300 |
| Nouns | Referent | −1.18 | .240 |
| | Operation | 0.29 | .772 |
| | Referent * Operation | 1.60 | .110 |
| Numbers | Referent | −0.01 | .994 |
| | Operation | 2.06 | .040 |
| | Referent * Operation | 0.79 | .429 |
| Key Terms | Referent | 0.27 | .787 |
| | Operation | −0.23 | .817 |
| | Referent * Operation | 0.05 | .961 |
| Non-key Terms | Referent | 0.37 | .710 |
| | Operation | −0.74 | .461 |
| | Referent * Operation | 1.07 | .286 |

**Table 8. Estimates of the final linear mixed model on Total Looking Times for the Target Referents. Final $p$ values calculated using the t-as-z method.**

**Fixed effects**

| | Est/Beta | SE | 95% CI | t | p |
|---|---|---|---|---|---|
| **Intercept** | 286.86 | 43.26 | 202.1–371.6 | 6.632 | <.001 |
| **Referent** | 95.02 | 29.27 | 37.66–152.4 | 3.247 | .001 |
| **Operation** | 127.75 | 45.97 | 37.65 −217.8 | 2.779 | .006 |
| **Referent * Operation** | −11.16 | 40.43 | −90.39–68.1 | −0.276 | .783 |

**Random effects**

| | Variance | | SD |
|---|---|---|---|
| **Participant (intercept)** | 40721 | | 201.8 |
| **Item (intercept)** | 10178 | | 100.9 |

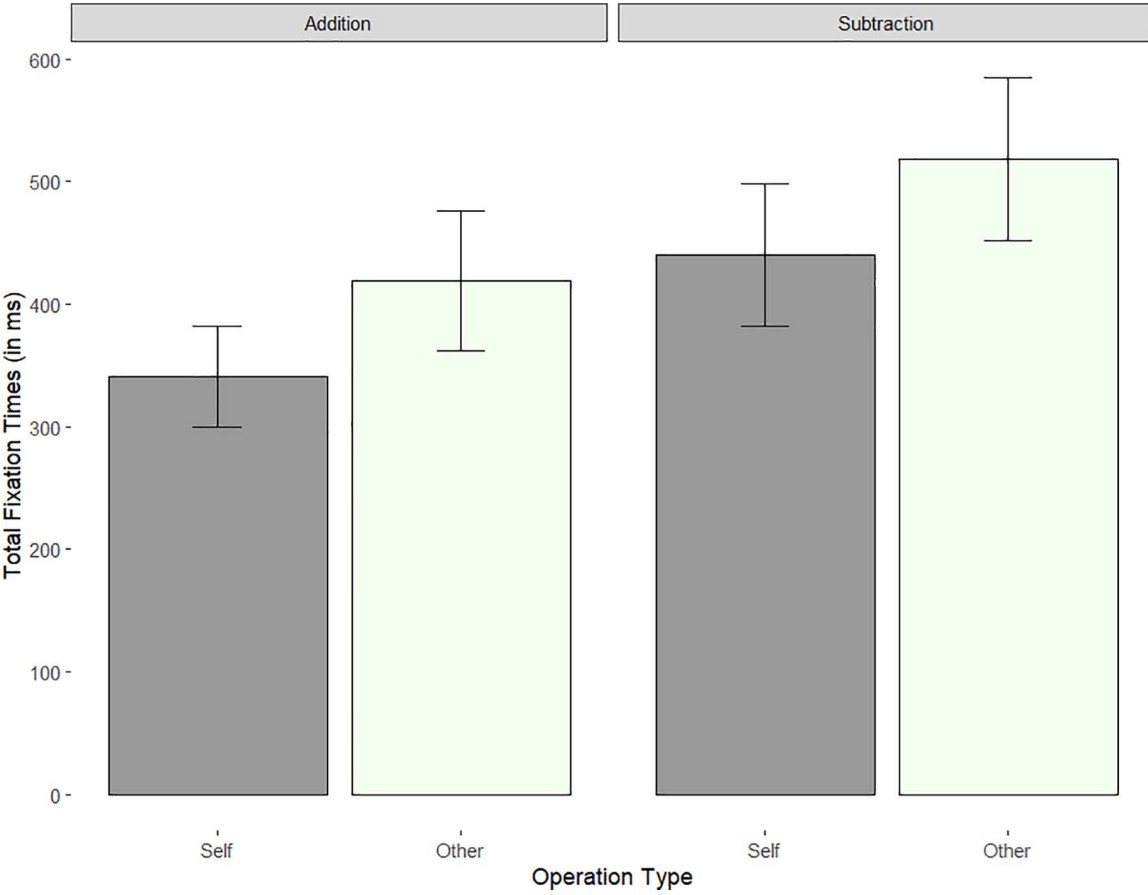

**Fig 3. Total fixation times on Target Referent words by Referent and Operation Type (bars indicate SE).**

there were no significant interactions between the two. For rigor, we conducted an identical exploratory analysis using proportional fixation times. Proportional fixation times on different word types were calculated by dividing the total fixations on each word type by the summed time on all word boxes. A similar set of mixed model analyses showed that again the effect of Referent was significant for the Referent Terms, with children spending a smaller proportion of their processing reading time on Self-referent Test pronouns ($M = 5.11\%$, $SE = .54$) than Other-referent Test pronouns ($M = 5.82\%$, $SE = .55$), $t = 3.4$, $p < .001$. No other effects were significant after corrections, all $ps > .035$.

## 4. Discussion

In the current study, we aimed to shed light on the underlying mechanism of the better performance observed in children when they solve self-referenced mathematical word problems as compared to other-referenced problems [12,15]. We presented children with age-appropriate numerical word problems and recorded their response times and eye fixations, hypothesising that word problem-solving time may be reflected in different fixation durations across referent conditions. We found significant effects of referent on response time and looking patterns, with self-referents producing quicker responses in subtraction problems, which are known to be more cognitively demanding [62]. Across both addition and subtraction problems, children spent significantly less time fixating on the pronoun itself (over one or multiple word visits), relative to other names/pronouns. This suggests that, rather than the introduction of self-referent terms drawing attention

and increasing task engagement, these terms may have a facilitatory effect on cognitive processing during numerical word problems, by reducing working memory demands.

The key contribution of the current study is a cogent explanation for the facilitative effects of self-pronouns in word-problem solving. It is possible that the decreased fixation time on personal pronouns is simply a consequence of reading facilitation: the word 'You' in self-referent problems is likely to be more familiar than other referent names, and children may therefore read it more quickly [63]. The fixation times for the 3-letter referent terms found in the current study (393ms for self-referent and 470ms for other-referent terms) can be contextualised by eye-tracking data from Gerth and Festman [64], who tested a similar sample of 10–12 year olds completing a close reading task. Mean first fixation times for short (3–5 letter) words in the close reading passages ranged from 208ms to 262ms, depending on children's general reading speed. The fixation durations for target referents in the current study were almost double this length, suggesting differences between referents represent revisitations rather than an initial reading time. Thus while discrepant reading times between the words 'You' and 'Zak'/'Eve' may account for some of the difference in fixation duration between them, the length of these fixations suggests that the effects of revisitation also require consideration.

Revisitations are a revealing measure because they tend to be higher in word problems with more complex numerical or linguistic processing demands [23,34], with the storage and processing of complex information putting a high load on working memory such that storage can fail. By promoting binding between active elements and facilitating activation of stored information, self-cues may allow problem information to be stored in working memory more easily [17,19]. For example, if the problem required a participant to remember that they had six apples, stored representations of themselves may make this construct more easily activated in working memory, with the item and anchoring referent term strongly bound. Thus, the participant would have less need to revisit this combination during operation processing; they may need to double-check the relevant number when performing the operation, but not require a reminder that they were the owner of the items in question, decreasing revisitation fixations on self-referent pronouns relative to other-referent names. Although our eye tracker did not have a high enough resolution to test this explanation by analysing the number of revisitations to specific words, a limitation that should be addressed by future research, the shorter total fixation duration on self-referent pronouns than other-referent names is consistent with the former requiring fewer revisitations during problem-solving.

A revisitation-based interpretation dovetails with the Situational Model of mathematical word-problem solving [21], with self-referenced problems facilitating the construction of the situational model relative to other-referenced problems by automatically binding elements of the propositional representations. This would lead to the need of fewer revisitations of the problem elements for self-referenced problems, and thus faster response times. The effect of Operation on the total fixation time is also consistent with the revisitation and situational model interpretation. Subtraction problems have a higher working memory load than those involving addition [62], so subtraction makes the construction of the situational model more difficult, perhaps eliciting more revisitation of all elements. Further, unlike previous studies [12,15], the current study did not find effects of self-referent cues on accuracy, with a high proportion of children performing at ceiling in their correct responses. This high performance level indicates that the required operations were within participants' routine word-problem solving capability, suggesting that differences in response and fixation time are likely to reflect the reading and problem interpretation involved in the development of the situational model (see [65]).

The current findings add to a growing literature that speaks to the effects of self-cues in applied contexts. Everyday tasks often involve a high working memory load, and more frequent use of self-cues such as personal pronouns in these tasks could be generally beneficial. This could include instructions and to-be-learned information being presented in the first person to support the binding of items, or more use being made of self-referent cues such as one's own name or face (see [66]). The effects of self-cues on mathematical problem-solving have now been demonstrated in both visual and verbal presentation (in the current study; [12,14,15]), and there is evidence of self-referent cues supporting the learning of new materials [13,67] and the ability to follow instructions [68]. The array of paradigms in which these effects have been

demonstrated suggests that at least in terms of tasks with a high working memory load, self-cue manipulations could be widely effective. The current study also adds to this evidence that the effects of self-cues on cognition are intrinsically linked with working memory storage, but more research is needed to understand how this varies across individuals [15,17].

In conclusion, the current study represents the first contribution of eye-tracking data to the study of self-cue effects on mathematical word-problem solving. Including self-referent terms increased word-problem solving speed in subtraction questions, and was associated with reduced time spent looking at the referent terms specifically (as compared with other character names) across both addition and subtraction problems. This finding may reflect the ability of self-cues to keep active and bind information in working memory, reducing the need to revisit them during word-problem solving. If the self-cue more generally facilitated problem solving by drawing attention and increasing task engagement, differences in visual attention to self and other referent terms within self-referent problems would not be expected. The study therefore provides a new theoretical insight into the application of self-reference effects in real-world tasks.

## Author contributions

**Conceptualization:** Janet F. McLean, Josephine Ross, Sheila J. Cunningham.

**Data curation:** Joshua March.

**Formal analysis:** Joshua March, Janet F. McLean, Jan R. Kuipers.

**Funding acquisition:** Janet F. McLean, Josephine Ross, Sheila J. Cunningham.

**Investigation:** Joshua March, Sheila J. Cunningham.

**Methodology:** Joshua March, Janet F. McLean, Josephine Ross, Sheila J. Cunningham.

**Project administration:** Janet F. McLean, Josephine Ross, Sheila J. Cunningham.

**Software:** Joshua March.

**Supervision:** Janet F. McLean, Josephine Ross, Sheila J. Cunningham.

**Validation:** Jan R. Kuipers, Sheila J. Cunningham.

**Visualization:** Joshua March.

**Writing – original draft:** Joshua March, Janet F. McLean, Josephine Ross, Jan R. Kuipers, Sheila J. Cunningham.

**Writing – review & editing:** Joshua March, Janet F. McLean, Josephine Ross, Jan R. Kuipers, Sheila J. Cunningham.

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
