## [Decision Letter · Decision Letter 0]

25 Nov 2025

PONE-D-25-48883Why do self-referent cues facilitate problem-solving? Insights from eye trackingPLOS ONE

Dear Dr. March,

Thank you for submitting your manuscript to PLOS ONE. After careful consideration, we feel that it has merit but does not fully meet PLOS ONE’s publication criteria as it currently stands. Therefore, we invite you to submit a revised version of the manuscript that addresses the points raised during the review process.

We look forward to receiving your revised manuscript.

Kind regards,

Laura Morett

Academic Editor

PLOS ONE

**Journal Requirements:**

“This work was funded by research grant ES/T000465/1 from the Economic and Social Research Council (ESRC).”

**Additional Editor Comments:**

I have now received two reviews for this manuscript, both of which are provide clear, detailed, and actionable feedback. Due to the high quality of the reviews, I will not reiterate their content here, but I encourage the authors to thoroughly address all of the points raised in a revision and a detailed response to reviewers. Provided the authors are willing to do so, I will attempt to send these documents to the same reviewers and ask them to evaluate the extent to which their concerns have been addressed to determine whether the revised manuscript is suitable for publication.

Reviewers' comments:

Reviewer's Responses to Questions

**Comments to the Author**

1. Is the manuscript technically sound, and do the data support the conclusions?

Reviewer #1: Yes

Reviewer #2: Partly

2. Has the statistical analysis been performed appropriately and rigorously? 

Reviewer #1: Yes

Reviewer #2: No

3. Have the authors made all data underlying the findings in their manuscript fully available?

The PLOS Data policy requires authors to make all data underlying the findings described in their manuscript fully available without restriction, with rare exception (please refer to the Data Availability Statement in the manuscript PDF file). The data should be provided as part of the manuscript or its supporting information, or deposited to a public repository. For example, in addition to summary statistics, the data points behind means, medians and variance measures should be available. If there are restrictions on publicly sharing data—e.g. participant privacy or use of data from a third party—those must be specified.requires authors to make all data underlying the findings described in their manuscript fully available without restriction, with rare exception (please refer to the Data Availability Statement in the manuscript PDF file). The data should be provided as part of the manuscript or its supporting information, or deposited to a public repository. For example, in addition to summary statistics, the data points behind means, medians and variance measures should be available. If there are restrictions on publicly sharing data—e.g. participant privacy or use of data from a third party—those must be specified.requires authors to make all data underlying the findings described in their manuscript fully available without restriction, with rare exception (please refer to the Data Availability Statement in the manuscript PDF file). The data should be provided as part of the manuscript or its supporting information, or deposited to a public repository. For example, in addition to summary statistics, the data points behind means, medians and variance measures should be available. If there are restrictions on publicly sharing data—e.g. participant privacy or use of data from a third party—those must be specified.requires authors to make all data underlying the findings described in their manuscript fully available without restriction, with rare exception (please refer to the Data Availability Statement in the manuscript PDF file). The data should be provided as part of the manuscript or its supporting information, or deposited to a public repository. For example, in addition to summary statistics, the data points behind means, medians and variance measures should be available. If there are restrictions on publicly sharing data—e.g. participant privacy or use of data from a third party—those must be specified.

Reviewer #1: Yes

Reviewer #2: Yes

4. Is the manuscript presented in an intelligible fashion and written in standard English?

Reviewer #1: Yes

Reviewer #2: Yes

5. Review Comments to the Author

Reviewer #1: This study used eye-tracking to investigate how self-referential pronouns influence children’s performance and processing during mathematical word problem solving. Although the valid sample size (N = 53) is relatively small, the experimental design is rigorous, and the discussion offers valuable insights. The following comments and suggestions are provided for the authors’ consideration.

1. During the eye-tracking procedure, was a chin rest used to stabilize participants’ heads? Please also report the sample rate of the valid eye-tracking data. Although the Tobii Pro X3-120 allows for some head movement, in my experience, the absence of a chin rest can result in considerable data loss.

2. In reading research, fixations shorter than 100 ms are typically removed. Did the present study follow this convention?

3. In Section 3.1 Data Cleaning, the sentence “Finally, we also removed trials for which the total fixation time on the word problem was < 1000 ms” is ambiguous. Does this refer to the total fixation duration across all areas of interest (i.e., all words) for a given problem?

4. Sections 3.2 and 3.3 report analyses based on a 2 (Referent) × 2 (Operation) design. It would be helpful to provide descriptive statistics (means and standard deviations) for the dependent variables—accuracy and response times—corresponding to each cell of the 2 × 2 design.

5. Table 2 is presented in an unusually simplified format. Please revise it according to the APA 7th edition guidelines for table presentation.

6. The number of decimal places across the manuscript is inconsistent. Please ensure uniform formatting in accordance with APA 7th edition requirements.

7. In Table 3, the mean value for Total Fixation Time of Non-Key Terms appears to be incorrect.

8. In Figure 3, the y-axis is labeled “Total Looking Time.” Is this intended to represent the same variable as “Total Fixation Time” mentioned in the text? Please ensure consistent terminology throughout.

Reviewer #2: Thank you for the possibility to review this very interesting contribution. The paper reports a study on the effect of self-referent cues (or personalization) on eye movements during word problem solving. The theoretical background of the paper is very sound and the rationale behind the research questions is flawless. The design of the study is also very well done and takes into accounts many of the methodological challenges of experimental eye-tracking research in a very effective way. However, starting with the reports of the results, the quality of the paper drops quite a bit. I will try to outline the issues that I see in the following. Due to the very promising design and the relevance of the topic, I am confident that the authors will be able to bring this manuscript into a version that is acceptable for publication after major revisions.

I only have two major remarks for the introduction:

- you should avoid the term “problem-solving“ and rather specify “word-problem solving” (p.12), as problem-solving usually refers to other tasks.

- p.13: There are some more recent works on eye movements and word problem solving which might be interesting (or not, please don’t see this as a request to cite these papers – it’s really just in case you are not aware and might find them helpful: https://doi.org/10.1037/cep0000366,
https://doi.org/10.1080/00461520.2019.1691004,
https://doi.org/10.1016/j.learninstruc.2025.102235)

Methods:

- If possible, you the calibration and validation accuracy would be very helpful to judge whether your eye-tracking setup was sufficiently accurate to distinguish the AOIs. I guess the words were far enough apart, but can you quantify that?

- Response time data cleaning (2.5 SD) – was this done with the log-transformed data? If not, would that make a difference?

- The way that the results of the GLMM and the LMM are reported seems unusual to me. The random effects structure itself is (to me) not as important as the parameter estimates. I think APA does not explicitly include GLLMs, but I personally think it is common to report a table with all fixed effects estimates, SE, CI, p, marginal and conditional R squared, and the random effects variances. I would ask the authors to do some research on best practice of reporting these analyses.

- In general, the presentation of the results is a bit hard to read. I can interpret the t-values, but only if I know how the predictors were coded. It would be helpful to also have a table with the descriptive mean values for each condition, or the estimated marginal means, or the fixed effects with a note on how the predictor was coded (I think one of the three would be sufficient).

Discussion

- The result that the pronouns are looked at shorter than the corresponding names. Interestingly, a total effect on response time was only reported for subtraction problems. So the effect of personalization is there, but why does it only affect the reaction time in subtraction problems? Your argument is that it is because the subtraction problems are harder, but this seems a bit far-fetched to me given that they were not actually solved less accurately in this study.

- p. 27 “reflected in different gaze patterns” – I would argue that fixation durations are not “gaze patterns” (in contrast to maybe revisits)

- p. 27 “relative to other character terms” – a) I don’t know what a “character term” is, but maybe this is a lack of English proficiency on my side b) The comparison was between the pronoun and the name, not between the Pronoun and the other parts of the problem, was it not?

- I don’t think the working memory hypothesis is supported by the data. In contrast, I think it even speaks against it: If working memory load had been increased by the use of impersonal names, this load should spill over to the reading of the other words as well. To me, the fact that only the fixation duration on the pronoun itself was quicker rather shows that this particular word is processed quicker (which is to be expected, since it is much more common than the corresponding names, as you also acknowledge). So while I think the hypothesis is absolutely valid, I think the data is not in favor of it. In contrast, I would argue that your data shows that it might be precisely the reading part (perception) that is affected, not the building of the situation model (which would be visible throughout the whole solution process and not just in fixations on the pronoun)

- “Did not have a high enough resolution to analyse the number of revisitations” – An eye tracker with a sampling rate of 8Hz should be more than capable of providing sufficient data to detect fixations. Taking only the raw data is not what this system is built for, and event detection algorithms (also open source ones) are very robust and efficient. Especially since the authors discuss the possible insights from analyses of revisits, I encourage them to find a way to run the data through an event-detection-algorithms and filtering for fixations. This would allow the authors to test the assumptions that they make in the discussion about revisits. (and this would also allow the authors to include the mean fixation duration as another outcome measure, which is often assumed to be related to working memory load).

6. PLOS authors have the option to publish the peer review history of their article (what does this mean?). If published, this will include your full peer review and any attached files.). If published, this will include your full peer review and any attached files.). If published, this will include your full peer review and any attached files.). If published, this will include your full peer review and any attached files.

...

Reviewer #1: No

Reviewer #2: No

---

## [Author Response · Author response to Decision Letter 1]

28 Jan 2026

[THE FOLLOWING IS ALSO INCLUDED AS A SEPARATE DOCUMENT IN OUR RESUBMISSION]

Thank you very much for the opportunity to revise and resubmit our article to PLOS One. We noted that the feedback from both reviewers was constructive and very supportive – we have now addressed their comments and feel that the paper is improved as a result. We would like to thank the reviewers for the time taken to review our manuscript, for their insight and for their support.

Below we show how we have addressed each comment, starting with the formatting comments from the Journal, then moving on to Reviewers 1 and 2. Thank you once again for your consideration of our manuscript. (N.B. all page numbers refer to the manuscript with tracked changes).

JOURNAL COMMENTS

https://journals.plos.org/plosone/s/file?id=ba62/PLOSOne_formatting_sample_title_authors_affiliations.pdf - We have now ensured we are naming the files correctly in our submission.

“This work was funded by research grant ES/T000465/1 from the Economic and Social Research Council (ESRC).”

Please include this amended Role of Funder statement in your cover letter; we will change the online submission form on your behalf.- We can confirm that the funders took no role in the study. Our updated Role of Funder statement should now read “This work was funded by research grant ES/T000465/1 from the Economic and Social Research Council (ESRC) – the funders had no role in study design, data collection and analysis, decision to publish, or preparation of the manuscript”.

3. Please note that funding information should not appear in any section or other areas of your manuscript. We will only publish funding information present in the Funding Statement section of the online submission form. Please remove any funding-related text from the manuscript. – We have now removed the funding information from our manuscript.

4. If the reviewer comments include a recommendation to cite specific previously published works, please review and evaluate these publications to determine whether they are relevant and should be cited. There is no requirement to cite these works unless the editor has indicated otherwise. – We have reviewed the papers suggested by one of the reviewers and included when relevant.

REVIEWER 1 COMMENTS

1. During the eye-tracking procedure, was a chin rest used to stabilize participants’ heads? Please also report the sample rate of the valid eye-tracking data. Although the Tobii Pro X3-120 allows for some head movement, in my experience, the absence of a chin rest can result in considerable data loss. - We thank the reviewer for this point – We thank the reviewer for this point - we have clarified that the participants’ heads were not secured on a chin rest throughout the study. We have also specified that the eye tracker samples eye data every 8 milliseconds on page 9.

2. In reading research, fixations shorter than 100 ms are typically removed. Did the present study follow this convention? - We did not remove fixations longer than 100ms as we didn’t calculate isolated fixations as individual events. As the data was very noisy (with lots of interrupted looking time on individual zones), we instead opted to sum up total looking times on the zones of interest.

3. In Section 3.1 Data Cleaning, the sentence “Finally, we also removed trials for which the total fixation time on the word problem was < 1000 ms” is ambiguous. Does this refer to the total fixation duration across all areas of interest (i.e., all words) for a given problem? - We thank the reviewer for this point – yes, this is what was meant and we have tried to make it clearer in our Data Cleaning section on page 16.

4. Sections 3.2 and 3.3 report analyses based on a 2 (Referent) × 2 (Operation) design. It would be helpful to provide descriptive statistics (means and standard deviations) for the dependent variables—accuracy and response times—corresponding to each cell of the 2 × 2 design. - We thank the reviewer for this suggestion – we have now included tables with the means / SDs for each of the levels of our variables, to accompany the results.

5. Table 2 is presented in an unusually simplified format. Please revise it according to the APA 7th edition guidelines for table presentation. - We thank the reviewer for this point – we have now provided a lot more information about our models for the reader, and have included tables suggested as best practice by Meteyard and Davis (2020) on pages 19, 21 and 26.

6. The number of decimal places across the manuscript is inconsistent. Please ensure uniform formatting in accordance with APA 7th edition requirements. - We have reviewed the manuscript to make sure we are consistent throughout for formatting.

7. In Table 3, the mean value for Total Fixation Time of Non-Key Terms appears to be incorrect. -We thank the review for catching this – we have fixed this error.

8. In Figure 3, the y-axis is labeled “Total Looking Time.” Is this intended to represent the same variable as “Total Fixation Time” mentioned in the text? Please ensure consistent terminology throughout. - We thank the reviewer for catching this – we have fixed this error and reviewed the rest of the figures for any additional inconsistency.

REVIEWER 2 COMMENTS

1. I only have two major remarks for the introduction:

- you should avoid the term „problem-solving“ and rather specify “word-problem solving” (p.12), as problem-solving usually refers to other tasks. – We have now used the more specific phrase mathematical (or numerical) word problem-solving throughout for clarity.

- p.13: There are some more recent works on eye movements and word problem solving which might be interesting (or not, please don’t see this as a request to cite these papers – it’s really just in case you are not aware and might find them helpful: https://doi.org/10.1037/cep0000366,
https://doi.org/10.1080/00461520.2019.1691004,
https://doi.org/10.1016/j.learninstruc.2025.102235) – Thank you for these suggestions for further reading, Two of the papers cited are relevant to the suggestion that self-referenced word problems may facilitate binding between numerical and word based aspects of the problem, we have added a note on page 6 that:

“This is consistent with eye tracking findings with young adults (Strohmaier et al., 2026) and 10- to 13-year-old children (Daroczy et al., 2025) which suggest that successful mathematical word problem solving relies on constructing a shared mental representation uniting text comprehension and numerical calculation.”

2. Methods:

- If possible, you the calibration and validation accuracy would be very helpful to judge whether your eye-tracking setup was sufficiently accurate to distinguish the AOIs. I guess the words were far enough apart, but can you quantify that ? - We specify that the word boxes varied from 6 to 9% screen width on page 12.

- Response time data cleaning (2.5 SD) – was this done with the log-transformed data? If not, would that make a difference? - We thank the reviewer for this suggestion – the initial cleaning was performed on the non-transformed response time data. We have investigated whether log-transforming the data first before cleaning it changes the results: the new cleaning method leads to only 5 trials difference between the datasets (1541 vs 1546) and we have confirmed that none of the findings are affected by the changes in dataset. The linear mixed models that converge are identical in structure, the significant findings remain significant for both the main analyses and follow-up tests, and no new tests are significant.

- The way that the results of the GLMM and the LMM are reported seems unusual to me. The random effects structure itself is (to me) not as important as the parameter estimates. I think APA does not explicitly include GLLMs, but I personally think it is common to report a table with all fixed effects estimates, SE, CI, p, marginal and conditional R squared, and the random effects variances. I would ask the authors to do some research on best practice of reporting these analyses. - We thank the reviewer for this feedback – we have researched how to best report the findings for mixed models to improve our results section. We now include tables inspired by best practice guidelines suggested by Meteyard and Davis (2020) on pages 19, 21 and 26, where we report as much information as possible for each model, to illustrate our findings alongside the random effects structure.

3. Results

- In general, the presentation of the results is a bit hard to read. I can interpret the t-values, but only if I know how the predictors were coded. It would be helpful to also have a table with the descriptive mean values for each condition, or the estimated marginal means, or the fixed effects[JM2] with a note on how the predictor was coded (I think one of the three would be sufficient). - We have now included the descriptive mean values for each condition in tables to accompany the results, and we have also explained how each predictor was coded, specifying the reference level for each variable.

4. Discussion

- The result that the pronouns are looked at shorter than the corresponding names. Interestingly, a total effect on response time was only reported for subtraction problems. So the effect of personalization is there, but why does it only affect the reaction time in subtraction problems? Your argument is that it is because the subtraction problems are harder, but this seems a bit far-fetched to me given that they were not actually solved less accurately in this study. - The reviewer makes the good point that the lack of an accuracy main effect means we do not currently justify our argument that subtraction problems were harder to solve. To address this issue, we have now expanded our description of ceiling effects in the accuracy data, breaking these down by operation to demonstrate difficulty differences. Specifically, we note that “ceiling effects were particularly pronounced in Addition questions, with 61.5% of participants having 100% accuracy across referent conditions (mean accuracy 91.9%). In Subtraction questions, a lower but still substantial 44.2% of participants had 100% accuracy (mean accuracy 86.5%). This suggests that while subtraction questions presented more of a challenge to participants, ceiling effects limited the sensitivity of the accuracy measure across operations” (p. 17). The inclusion of this descriptive ceiling data in the Results has also allowed us to be more specific in the Discussion, noting that there was a “a high proportion of children performing at ceiling in their correct responses” (p. 29).

- p. 27 “reflected in different gaze patterns” – I would argue that fixation durations are not “gaze patterns” (in contrast to maybe revisits) – replace w fixation durations - We agree and have replaced this term with “fixation durations”.

- p. 27 “relative to other character terms” – a) I don’t know what a “character term” is, but maybe this is a lack of English proficiency on my side b) The comparison was between the pronoun and the name, not between the Pronoun and the other parts of the problem, was it not? – replace character term with name - We agree here and have replaced with “name” to improve clarity.

- I don’t think the working memory hypothesis is supported by the data. In contrast, I think it even speaks against it: If working memory load had been increased by the use of impersonal names, this load should spill over to the reading of the other words as well. To me, the fact that only the fixation duration on the pronoun itself was quicker rather shows that this particular word is processed quicker (which is to be expected, since it is much more common than the corresponding names, as you also acknowledge). So while I think the hypothesis is absolutely valid, I think the data is not in favor of it. In contrast, I would argue that your data shows that it might be precisely the reading part (perception) that is affected, not the building of the situation model (which would be visible throughout the whole solution process and not just in fixations on the pronoun) - The reviewer suggests that faster reading of the self-referent word ‘You’ than the other referent name (Zak or Eve) provides a more compelling explanation for the difference in fixation time between these words than our working memory hypothesis. This explanation was mentioned briefly in our original manuscript, but we now discuss it in more detail and in the context of research on typical short word fixation times. These times suggest children in the current study are revisiting the referent terms to aid their problem-solving, justifying our consideration of the role of revisitation. Thus we note that “while discrepant reading times between the words ‘You’ and ‘Zak’/‘Eve’ may account for some of the difference in fixation duration between them, the length of these fixations suggests that the effects of revisitation also require consideration” (p. 28).

We now consider the revisitation/working memory explanation in a separate paragraph, explaining in more detail why this provides an account of fixation duration differences between ‘You’ and ‘Zak’/‘Eve’ specifically, rather than spilling over to other words in the problem. We illustrate this with an example, noting that children “they may need to double-check the relevant number when performing the operation, but not require a reminder that they were the owner of the items in question, decreasing revisitation fixations on self-referent pronouns relative to other-referent names” (p. 29). However, we finish the paragraph with a note of caution, commenting that the limitations of our eye tracker meant that we could not test this explanation by analysing the number of revisitations and suggesting that this should be addressed by future research.

- “Did not have a high enough resolution to analyse the number of revisitations” – An eye tracker with a sampling rate of 8Hz should be more than capable of providing sufficient data to detect fixations. Taking only the raw data is not what this system is built for, and event detection algorithms (also open source ones) are very robust and efficient. Especially since the authors discuss the possible insights from analyses of revisits, I encourage them to find a way to run the data through an event-detection-algorithms and filtering for fixations. This would allow the authors to test the assumptions that they make in the discussion about revisits. (and this would also allow the authors to include the mean fixation duration as another outcome measure, which is often assumed to be related to working memory load). - We agree with the reviewer that event detection algorithms are increasingly used in the eye tracking literature. However, in our pre-registration we detailed an analysis and we stuck as much as we could with these methods. We chose not to try to calculate revisitations, with whichever method, because visual inspection of the data revealed that revisitations were particularly difficult to establish due to the high noise levels in the data. Then, although highly successful, event detection algorithms are not protected against noise-driven backward movements which look like regressions. Assuming that noise is randomly distributed around fixation, upward and downward noise will be cancelled out in the algorithms, but right, and leftward noise can be classified as regressions by the model. Therefore, while using these algorithms and depending on the settings, words that are fixated most, will typically lead to the most regressions, introducing a bias in the output. We therefore

---

## [Decision Letter · Decision Letter 1]

24 Mar 2026

Why do self-referent cues facilitate problem-solving? Insights from eye tracking

PONE-D-25-48883R1

Dear Dr. March,

We’re pleased to inform you that your manuscript has been judged scientifically suitable for publication and will be formally accepted for publication once it meets all outstanding technical requirements.

Kind regards,

Laura Morett

Academic Editor

PLOS One

Additional Editor Comments (optional):

Apologies again for the protracted review process due to difficulties securing two reviewers with relevant expertise initially. At this point, the reviewers are satisfied with the revised manuscript, and I agree that it is suitable for publication. Therefore, I am, pleased to recommend acceptance.

Reviewers' comments:

Reviewer's Responses to Questions

**Comments to the Author**

1. If the authors have adequately addressed your comments raised in a previous round of review and you feel that this manuscript is now acceptable for publication, you may indicate that here to bypass the “Comments to the Author” section, enter your conflict of interest statement in the “Confidential to Editor” section, and submit your "Accept" recommendation.

Reviewer #1: All comments have been addressed

Reviewer #2: All comments have been addressed

2. Is the manuscript technically sound, and do the data support the conclusions?

Reviewer #1: Yes

Reviewer #2: Yes

3. Has the statistical analysis been performed appropriately and rigorously? 

Reviewer #1: Yes

Reviewer #2: Yes

4. Have the authors made all data underlying the findings in their manuscript fully available?

The PLOS Data policy requires authors to make all data underlying the findings described in their manuscript fully available without restriction, with rare exception (please refer to the Data Availability Statement in the manuscript PDF file). The data should be provided as part of the manuscript or its supporting information, or deposited to a public repository. For example, in addition to summary statistics, the data points behind means, medians and variance measures should be available. If there are restrictions on publicly sharing data—e.g. participant privacy or use of data from a third party—those must be specified.requires authors to make all data underlying the findings described in their manuscript fully available without restriction, with rare exception (please refer to the Data Availability Statement in the manuscript PDF file). The data should be provided as part of the manuscript or its supporting information, or deposited to a public repository. For example, in addition to summary statistics, the data points behind means, medians and variance measures should be available. If there are restrictions on publicly sharing data—e.g. participant privacy or use of data from a third party—those must be specified.requires authors to make all data underlying the findings described in their manuscript fully available without restriction, with rare exception (please refer to the Data Availability Statement in the manuscript PDF file). The data should be provided as part of the manuscript or its supporting information, or deposited to a public repository. For example, in addition to summary statistics, the data points behind means, medians and variance measures should be available. If there are restrictions on publicly sharing data—e.g. participant privacy or use of data from a third party—those must be specified.requires authors to make all data underlying the findings described in their manuscript fully available without restriction, with rare exception (please refer to the Data Availability Statement in the manuscript PDF file). The data should be provided as part of the manuscript or its supporting information, or deposited to a public repository. For example, in addition to summary statistics, the data points behind means, medians and variance measures should be available. If there are restrictions on publicly sharing data—e.g. participant privacy or use of data from a third party—those must be specified.

Reviewer #1: Yes

Reviewer #2: No

5. Is the manuscript presented in an intelligible fashion and written in standard English?

Reviewer #1: Yes

Reviewer #2: Yes

6. Review Comments to the Author

Reviewer #1: The authors have addressed all the concerns raised in the previous round of review. The revisions are appropriate, and the clarifications provided have significantly improved the clarity of the manuscript. I have no further comments and recommend the paper for publication.

Reviewer #2: Thank you for the opportunity to review this paper again. The authors have taken all previous comments seriously and provided detailed responses. I think the manuscript improved substantially, and I think it can be accepted in this form. I have some smaller remaining comments for the authors to consider, but none of these points require an additional round of reviews from my end.

I have now (in contrast to my first review) clicked the box that not all data is freely available. This is because I wanted to check whether indeed the data quality is insufficient to run an event detection algorithm. I noticed that only the aggregated eye-tracking data is uploaded, but not the raw data. The editor should decide if this is necessary.

In my first review, I noted that the SD-cutoff was done before the log-transformation. The authors have convincingly shown that this does not make a difference in the interpretation of the data and thus decided to go with their original analysis. Similarly, they have decided not to include the suggested analyses of revisits, party with the argument that they want to stick as much as they can with the pre-registration. Honestly, this seems a bit like the easy answer. The argument that the authors want to follow the re-registration as closely as possible seems to contradict the purpose of the peer-review process: If the analysis is not open for debate, there is no use in a review process. The fact that the log-transformation does not make a difference does not mean that it is the better way to do it. I would have expected the authors to evaluate which approach is sounder (on a theoretical basis), and then choose this approach (whichever it is), rather than simply checking if changes are necessary and then stick with their previous version.

I would still argue that the sampling rate of 8 Hz is high enough to analyze revisits. If data quality is indeed the bottleneck, this should be made explicit in the discussion instead of arguing that the sampling rate was insufficient. I also don’t think it is correct that event detection algorithms only filter upwards and downwards movements. Again, these arguments sound a bit like an excuse not to change the analyses, which is understandable, but should not be the guiding principle.

A small detail: Fixation count should be multiplied by 8 1/3, not by 8, since 1000/120 is not 8. This might be the ultimate test: Implementing this will not change any results but will be annoying and correct. I call for the authors’ perfectionism and thank them for allowing me (without a choice) to formulate my critique so openly. Thanks for the great work and the productive process, all the best with the publication!

7. PLOS authors have the option to publish the peer review history of their article (what does this mean?). If published, this will include your full peer review and any attached files.). If published, this will include your full peer review and any attached files.). If published, this will include your full peer review and any attached files.). If published, this will include your full peer review and any attached files.

...

Reviewer #1: **Yes:** Chao-Jung WuChao-Jung WuChao-Jung WuChao-Jung Wu

Reviewer #2: No

---

## [Editor Report · Acceptance letter]

PONE-D-25-48883R1

PLOS One

Dear Dr. March,

I'm pleased to inform you that your manuscript has been deemed suitable for publication in PLOS One. Congratulations! Your manuscript is now being handed over to our production team.

Kind regards,

on behalf of

Dr. Laura Morett

Academic Editor

PLOS One